# An Untargeted Metabolomics Approach on Carfilzomib-Induced Nephrotoxicity

**DOI:** 10.3390/molecules27227929

**Published:** 2022-11-16

**Authors:** Ioanna Barla, Panagiotis Efentakis, Sofia Lamprou, Maria Gavriatopoulou, Meletios-Athanasios Dimopoulos, Evangelos Terpos, Ioanna Andreadou, Nikolaos Thomaidis, Evangelos Gikas

**Affiliations:** 1Laboratory of Analytical Chemistry, Department of Chemistry, National and Kapodistrian University of Athens, 15771 Athens, Greece; 2Laboratory of Pharmacology, Department of Pharmacy, National and Kapodistrian University of Athens, 15771 Athens, Greece; 3School of Medicine, Department of Clinical Therapeutics, National and Kapodistrian University of Athens, 11527 Athens, Greece

**Keywords:** carfilzomib, nephrotoxicity, metabolomics, HRMS

## Abstract

Background: Carfilzomib (Cfz) is an anti-cancer drug related to cardiorenal adverse events, with cardiovascular and renal complications limiting its clinical use. Despite the important progress concerning the discovery of the underlying causes of Cfz-induced nephrotoxicity, the molecular/biochemical background is still not well clarified. Furthermore, the number of metabolomics-based studies concerning Cfz-induced nephrotoxicity is limited. Methods: A metabolomics UPLC–HRMS–DIA methodology was applied to three bio-sample types i.e., plasma, kidney, and urine, obtained from two groups of mice, namely (i) Cfz (8 mg Cfz/ kg) and (ii) Control (0.9% NaCl) (*n* = 6 per group). Statistical analysis, involving univariate and multivariate tools, was applied for biomarker detection. Furthermore, a sub-study was developed, aiming to estimate metabolites’ correlation among bio-samples, and to enlighten potential mechanisms. Results: Cfz mostly affects the kidneys and urine metabolome. Fifty-four statistically important metabolites were discovered, and some of them have already been related to renal diseases. Furthermore, the correlations between bio-samples revealed patterns of metabolome alterations due to Cfz. Conclusions: Cfz causes metabolite retention in kidney and dysregulates (up and down) several metabolites associated with the occurrence of inflammation and oxidative stress.

## 1. Introduction

Carfilzomib (Cfz) is a second-generation proteasome inhibitor, licensed for the curation of relapsed/refectory multiple myeloma. The Cfz drug is distributed to all human tissues except the brain and is rapidly degraded by peptidase cleavage and epoxide hydrolysis, providing non-active metabolites. The metabolism of Cfz is extra-hepatic and independent of the liver function [1]. Despite Cfz’s therapeutic activity, the drug has been associated with cardiorenal side effects. However, the interplay and patho-mechanism of Cfz’s adverse effects are still elusive. Noteworthily, some preclinical studies have presented potential prophylactic therapies against Cfz’s induced cardiorenal complications, but none of them have been applied into the clinical practice. Therefore, elucidation of the underlying mechanisms of Cfz’s cardiorenal effects is an unmet clinical need. Moreover, clinical practice is derived from biomarkers and diagnostic tools that can detect the early onset of Cfz-related cardio-renal complications, and this is an additional need of the hematologic research [1,2,3,4].

Concerning nephrotoxicity, several cases of renal failure have been reported, although with no defined hematologic evidence [5]. According to Fotiou et al., the effects of Cfz in kidneys could be described by two potential mechanisms; the first involves the Cfz-induced microvascular toxicity to the renal endothelium, and the second involves the hyperactivation of the complement membrane attack complex [5]. Therefore, an increase in plasma creatinine was reported as an indicator of renal failure.

In an effort to elucidate the mediators of Cfz’s nephrotoxicity, metabolomics is a powerful weapon. The metabolome is a sensitive recipient of every (internal and external) perturbation, and the study of its alterations is an efficient way to detect and/or describe several pathological conditions, or even their progress. There are two main approaches to metabolome’s investigation, as follows: (i) in cases where there is *a priori* knowledge of a condition’s metabolic background, targeted metabolomics can determine specific metabolites as potential biomarkers, or (ii) in hypothesis-free cases, the exploratory analysis of the whole metabolome (untargeted metabolomics) could provide significant evidence, enlightening researchers as to the underlying biochemical causes [6,7,8]. Thus, the untargeted metabolomics approach can reveal hidden aspects of Cfz’s related renal dysfunction, as the biochemical mechanism is still non clarified. The existing metabolomics studies on both Cfz nephrotoxicity and cardiotoxicity are still limited. Tantawy et al., have performed the only a multi-omics study, including the plasma untargeted metabolomics as well as focusing on Cfz-induced cardiotoxicity [9].

Our recent targeted metabolomics study was the first report that linked Cfz with renal injury. The results showed extended dysregulation of renal metabolism due to the administration of the drug [10]. Thus, in order to further investigate Cfz’s impact on cardiorenal related metabolism, a complementary untargeted metabolomics approach was employed. The in vivo experimentation was performed by the administration of a translational dose, equivalent to the human therapeutic one, which is previously established to mimic the clinically observed cardiorenal adverse effects in mice [1]. An MS-based methodology was selected to ensure the high sensitivity in metabolite detection. Furthermore, the experimental set up was carefully optimized to the needs of the untargeted metabolomics approach, in order to investigate the “holistic picture” of metabolic alterations. The impact of Cfz was examined in plasma, kidneys, and urine metabolome, aiming to determine how the drug affects the circulatory system. For this reason, the kidneys have been considered as an input/output system between blood and urine excretion. An intriguing aspect of the developed statistical methodology was to reveal relations of the differentially regulated biomarkers that were discovered in the three biomaterials, to uncover inter-circulatory metabolic correlations. Furthermore, attempting to identify as many metabolites as possible, the identification procedure considered potential products of metabolites’ metabolism as well, under the notion of the “dark metabolome” [11].

## 2. Results

### 2.1. Data Pre-Processing

Data independent acquisition (DIA) is commonly adopted for untargeted metabolomics, offering both quantitative and structural information by alternately recording low collision energy MS and high collision energy MS (fragmentation spectra) [12]. Despite DIA’s advantages, the facile and consistent interpretation of the vast amount of generated data remains challenging. Among a large number of peak picking software, MZmine version 2.51 [13] was selected, as its modular nature provides the ability to evaluate every step of the process while offering the ability to select between a large variety of advanced tools.

The separation of the low and high collision energy (CE) MS spectra was a crucial step for the rest of the procedure, as these two types provide different challenges. The low CE–MS spectra provide information about the intact metabolite structure, while they are also reproducible enough for semi-quantitation. Low CE–MS information is intended for the biomarkers’ determination, so the removal of noise or of non-reproducible peaks is important for the further chemometric analysis, whereas high CE–MS, which are intended to facilitate the identification, are “fuzzier”, with lower signal intensity and mass accuracy. Thus, a peak-picking protocol with common settings for both low and high CE–MS would not be efficient. In the peak-picking procedure, the removal of isotopes and adducts, the chromatograms’ normalization, the alignment according to retention time (t_R_)_,_ by employing “well behaved” peaks as standards [14], and the final correction of the aligned list removing the duplicates and gap-filling using the bellow referring algorithms were performed. For the above steps, some parameters were universally set in all datasets and others were adjusted according to the bio-sample type and the ionization mode.

Signal-intensity correction was the penultimate step before conducting the statistical analysis because this study, similar to all long-term metabolomics studies, “suffered” from non-negligible drift of signal intensity attributed to several factors, such as the non-reproducible ESI-LC-MS system, or detector performance [15,16]. To overcome this issue, QC samples were periodically analysed between samples in such a way that three samples were “blocked” by two QC samples. Subsequently, after peak-picking, statTarget2 [17] (utilizing QCs and the QC-RLSC algorithm which employs the LOWESS function), was used to normalize the signal among samples [18]. The results of peak-picking and QC-RLSC signal correction are summarized in Table 1.

### 2.2. Statistical Analysis

A combination of multivariate and univariate analysis was performed for the determination of the variables with the higher contribution to group discrimination. Multivariate analysis was performed using SIMCA 14.1 (Umetrics, Sweden, Upsala). Variables with values which did not differ from the median were excluded before the analysis. The number of features included in multivariate analysis is described at Table 1. Then, PCA analysis was employed to investigate any discrimination trends and possible outliers between the Cfz and Control samples, while PLS-DA models were developed to determine the most differentiative features. Permutation testing was employed to estimate overfitting concerning the Q^2^ value. It should be noted that the prediction ability is limited but that is justified due to ethical reasons for the use of lab animals. The results of all datasets’ PLS-DA models are summarized in Table 2. The observations that emerged from the multivariate statistical analysis revealed Cfz’s influence on all bio-sample types. Referring to kidney-PCA score plots (Figure 1), the two groups are well defined and there is a clear separation between the Cfz and Control samples. In urine, there is a separation tendency, more apparent in the negative mode; however, there is a significant dispersion among samples of the same group. In plasma, the discrimination is quite fuzzy. The PCA score plots suggested that the main effect of Cfz’s administration is located in the renal tissue and is partially expressed in urine excretion as well, whereas the impact on plasma is quite vague. The PCA plots from the positive ionization datasets are presented in Figure 1 and the plots of the negative ionization have been attached in the Appendix A.

The PLS-DA analysis succeeded in achieving classification between the Cfz group and the Control group for all datasets except plasma (−) and, therefore, the latter was not considered for the feature selection. The kidney showed the highest number of differentiating variables (more VIP values > 1). In an effort to render the analysis more rigorous, only the variables with VIP score higher than 1.5 were considered as potential biomarkers, as they were supposed to exert a higher impact on a group’s classification and were submitted for further investigation. A summary of the PLS-DA results, as well as those from the models’ validation, are provided in Table 2**.** The PLS-DA score plots of positive ionization are presented in Figure 1**,** while those from the negative are given in the Appendix A.

Additionally, univariate analysis was implemented, employing MetaboAnalyst 5.0 [19] for ROC curve analysis, an FDR-corrected *t*-test (tt), fold change (FC) analysis, and the creation of volcano plots. The results obtained suggested that the most differentiating variables were detected in the kidneys, which is in accordance with the multivariate analysis. Only the variables with AUC values higher than 0.9 were selected for identification. Therefore, it has been postulated that the main differentiation due to Cfz’s administration is expressed in the kidney metabolome and is also expressed in the urine metabolites. However, neither Cfz nor its administration effects on renal regulation seem to be expressed in blood, as its metabolic composition is essentially not influenced. This may lead to an early conclusion that the effect of the drug is focused on the renal function.

Regarding the steps of the data treatment methodology, the large number of differentiated features resulted from both univariate and multivariate analysis and was limited based on AUC (>0.9) and VIP (>1.5) values; only those were submitted for identification.

### 2.3. Peaks Identification

Concerning DIA, the attribution of peaks to metabolites is a laborious task. As described above, the high CE–MS were used to assign structural features assisting in the unequivocal peaks’ identification. However, although this information exists in the high CE–MS, it cannot be accurately related to a specific precursor ion, since during the bbCID scan mode all precursors are fragmented simultaneously. Thus, the most discriminant features of each dataset (VIP > 1.5, AUC > 0.9) were individually extracted as ion chromatograms (IC) using the DataAnalysis software (Bruker Daltonics, Bremen, Germany) and the background spectra were removed. The cleaned bbCID spectra of the IC were considered as a type of “pseudo-MSMS”, under the notion that high CE–MS info was obtained in a firm retention time range, strictly correlated to the retention time of a low CE–MS spectrum. As many of the discriminant variables could not be identified as metabolites, the experimental pseudo-MSMS was searched alongside the HMDB 5.0 database [20], working towards its implementation in the MyCompoundID (MCID) online library (University of Alberta), http://www.mycompoundid.org/mycompoundid_IsoMS/ (accessed on 20 June 2022). In addition to the human endogenous metabolites, the MCID includes their predicted metabolic products as well. The identification procedure involved 76 features, of which 54 were finally identified. The results concerning plasma (+) showed the lowest number of identified metabolites, and the most differentiating features were considered as formate adducts. Applying the identification workflow in the first four most differentiating features (i.e those with the higher VIP values) of the plasma (+) dataset, it was observed that their mass spectra were identical to those of the calibrant solution, as shown in Figure 2. This is attributed to the endogenous formate which, under the analysis conditions, showed the same MS as the calibrant solution, but was chromatographed as well, showing a t_R_ at 7.16 min. The total summary of the most differentiating metabolites of the Cfz and Control groups are represented on Table 3.

A total of 71% of the features were annotated, with 75% of them being detected as products of metabolites’ metabolism.

The majority of the identified metabolites are primary or secondary endogenous metabolites, whereas only two of them, *6-perillyl alcohol* and *3-hydroxy-n-methylcarnitine*, belong to nutrients. The last observation may also suggest some differentiation in digestion due to Cfz administration.

## 3. Discussion

During the identification procedure, it was observed that a significant number of features could not be attributed to already known metabolites. Nevertheless, those features did represent real and well-behaved peaks with reproducible signal, mass accuracy, and retention time. Furthermore, they belonged to the most discriminative variables and, thus, their existence could not be ignored as they may belong to the “dark metabolome” [11]. Aiming to annotate more of those features to metabolites, the products of metabolites’ one reaction metabolism were also searched as potential renderings through the respective module of the MCID library. This attempt provided associations of the given *m*/*z* (and their MS2 thereof) with metabolites that have been submitted to endogenous enzymatic addition or loss reactions of chemical groups, known to be involved in metabolism.

Several metabolites that are already correlated with cardiovascular diseases and renal disorders were detected as differentially regulated compounds in the kidney and urine samples of the Cfz group.

### 3.1. Asymetric Dimethylarginine

Asymmetric dimethylarginine (ADMA) has been detected in plasma, kidney, and urine samples in three compound forms (*ADMA, ADMA + SO_3_, ADMA + C_5_H_4_N_2_O*). The ADMA is an endogenous metabolite, existing in plasma and tissues, whereas it is appearing in urine as a metabolic product. The compound is produced during protein methylation in the presence of arginine’s residues, by protein arginine methyltransferases (PRMTs) [21]. Furthermore, ADMA acts through the inhibition of nitric oxide (NO) production, competing with l-arginine in binding to the active site of nitric oxide synthase (NOS) enzymes [22], and resulting to reduction in NO bioavailability. The NO produced in endothelial cells acts as vasodilator and as anti-atherogenic agent due to its anti-inflammatory and anti-thrombotic activity. The interference of NO synthesis invokes dysregulation of endothelium vascular homeostasis [23], whereas the resulting reduction in NO levels elevates blood pressure and renal vascular resistance. Thus, the increase in ADMA circulating plasma levels is associated with cardiovascular and renal diseases [24]. Therefore, ADMA is considered as a marker of chronic kidney disease or cardiovascular disease. Interestingly, in the current study, the levels of ADMA in plasma were not statistically different between the two groups but appeared differentiated in the kidney samples, where ADMA was found to be almost two-fold increase in the Cfz group. The increased kidney levels of ADMA imply the increased expression or action of PRMTs or is a sign of elevation of renal metabolic rate. Those observations may describe an instant and probably temporal effect of Cfz on renal tissue, responsible for ADMA’s increase and subsequently for the reduction in NO synthesis. This reduction may induce a kind of vascular damage or inflammation and, consequently, acute kidney injury [25].

Mice with ischemia/reperfusion injury showed high renal levels of ADMA [26] and, therefore, the compound may be a marker of this condition. The above study also related ADMA’s renal levels with oxidative stress, as the compound was correlated with 8-hydroxy-2′-deoxyguanosine, a marker of oxidative stress.

### 3.2. N1-Methyl-2-pyridone-5-carboxamide

Here, N1-Methyl-2-pyridone-5-carboxamide (2PY) is an end-product of nicotinamide metabolism. The compound has been already associated with uremia and chronic kidney disease (CKD) and is registered as a uremic toxin. The toxicity of 2PY is related with compound’s inhibiting activity against poly (ADP-ribose) polymerase-1 (PARP-1) [27]. The PARP-1 participates in several mechanisms, such as differentiation and proliferation, DNA damage repair through chromatin’s reshaping, and in the regulation of inflammation, providing cell death or inducing the migration of leukocytes under several conditions. Moreover, PARP-1 is necessary for inducible nitric oxide synthase (iNOS), which promotes NO production [28,29]. Analogously to the ADMA-case, the detection of 2PY implies the reduction in NO production. However, in this case, high levels of 2PY were detected only in the urine of Cfz mice, as the kidney levels of Cfz mice presented a slight increase and their plasma levels were statistically equal. A recent study revealed the correlation between normal renal function and 2PY excretion in urine, namely that the levels of 2PY in patients with kidney damage (renal transplant recipients) were elevated compared to the respective levels of healthy donors. Additionally, the levels of the healthy donors were increased after a kidney donation operation [30]. In the case of Cfz, the increased levels of 2PY in the urine of treated mice may imply the induction of a locally estimated effect of renal function on the urinary system.

### 3.3. N4-Acetylcytidine

The elevation of N4-acetylcytidine (ac4C) levels in the urine of treated mice was also observed. Here, ac4C is an endogenous nucleoside, a urinary product of RNA catabolism, produced by the action of N-acetyltransferase 10 (NAT10). The urine and blood ac4C levels have been associated with several diseases. According to Jin G. et al., an increased level of ac4C in urine is a sign of inflammatory response. This, combined with the elevation of other modified nucleosides, is observed in patients with uremia. However, ac4C levels in patients with chronic renal failure are decreased. Thus, it is assumed that potential abnormalities of RNA degradation induce irregular accumulation of ac4C in uremic patients. In addition, high levels of ac4C are reported in hypertensive rats. Finally, ac4C increase is associated with oxidative stress in eukaryotes but it is not yet clear if the ac4C increase in urine is the result of this condition [31].

### 3.4. Phenylacetic Acid

Phenylacetic acid (PAA) is a registered uremic toxin, detected in high levels in the urine of Cfz mice. The PAA is a product of phenylalanine’s catabolism, increased in the blood of patients with chronic kidney disease and uremia. Furthermore, PAA is an inhibitor of iNOS expression, such as 2PY, and inhibits plasma membrane calcium ATPase. Thus, PAA is suggested to participate in artery reshaping [32]. Additionally, as a uremic toxin, PAA is involved in the activation of polymorphonuclear leucocytes (PMNLs) inducing inflammation. This observation has been verified by in vitro experiments that point out PAA’s contribution in the inflammation induction and in the decline of PMNL apoptosis [33].

### 3.5. 2-Aminoisobutyric Acid

The 2-Aminoisobutyric acid is an amino acid that has been found to be decreased in the kidneys of treated mice. This compound is reported to prevent kidney tubulopathy, as it inhibits the action of D-serine. The D-isomer of serine is a nephrotoxic agent, causing selective necrosis of the S3 segments of proximal tubules. The D-serine is reabsorbed through the proximal convoluted tubule of kidney and is degraded, at the same location by the enzyme D-amino acid oxidase (d-AAO), into the corresponding a-keto acid and ammonia. Moreover, the catabolism of D-serine induces the generation of H_2_O_2_ and is assumed to lead to a decrease in renal cellular glutathione, resulting in high production of ROS and oxidative stress. As the structure of 2-aminoisobutyric acid corresponds to that of D-serine, it is possible that the presence 2-aminoisobutyric acid in the kidney prevents the oxidative stress inhibiting D-serine’s catabolism [34,35]. Considering that the decreased levels of 2-aminoisobutiric acid may be involved in kidney injury, provoked by limited inhibition of D-serine catabolism, the correlation between these two compounds in the Cfz and Control kidney mice was studied. The outcome was the detection of increased D-serine levels in Cfz mice, while the compound was not detected in the Control samples, as is represented in Figure 3. The correlation of D-serine and 2-aminoisobutyric acid is also represented in heatmap in the Appendix A. Therefore, the hypothesis that kidney injury caused by oxidative stress conditions induced by the decreased levels of 2-aminoisobutyric acid is verified.

As a general observation, the outcome of the untargeted metabolomics study has revealed several other metabolites that have already been related to renal dysfunction diseases, besides of those described above. These are as follows: galactonic acid has been detected as a biomarker of CKD [36], threonic acid has been related to oxidative stress induction in patients with membranous nephropathy [37], 2,3-diaminopropionic acid is associated with epithelial cell necrosis of the proximal straight tubules, such as D-serine [38], and octanoylcarnitine, 2-hexenoylcarnitine, and valerylcarnitine, as members of acylcarnitines, are also correlated with AKI [39]. Deoxyinosine is also a nephrotoxicity biomarker [40], whereas indoxyl glucuronide has already been detected in the biofluids of uremic patients [41]. Moreover, some metabolites show differentiation of nutrient’s metabolism between two groups which may act on renal function, such as methylxanthine [42].

### 3.6. Exploration of Metabolites Alterations between Different Bio-Samples

As mentioned above, the metabolic profile of kidney and urine datasets seem to be more affected by Cfz administration, in contrast to plasma. In order to explore the potential correlation of metabolites among the kidney as an input/output system, the most discriminant metabolites and their metabolic products were used as “targeted substances” and were semi-quantitated in all the examined bio-samples. The estimated peak areas of the detected metabolites were used to plot the mean metabolite content in every type of bio-sample, along with the standard deviation of the group. According to the observed patterns, the metabolites can be categorized in five potential patterns depending on their variation among plasma, the kidneys, and urine, as follows: (a) metabolites detected in all samples and differentiated only in the kidneys, (b) metabolites detected in all systems and differentiated only in urine, (c) metabolites detected in all systems and differentiated in both the kidneys and urine, (d) metabolites detected and differentiated only in the kidneys, and I metabolites detected and differentiated only in urine. Examples of the different plot patterns are shown in Figure 4. The plasma differences of treated and Control samples were not significant. In addition, the metabolites detected in all samples, were, in the majority of cases, increased in the kidneys of Cfz group. Furthermore, most metabolites are increased in the kidneys and urine of Cfz mice. This observation fosters the idea that Cfz administration increases the renal metabolism of mice. This demands higher consumption of oxygen, a fact that causes hypoxia in parts of the kidney or affects blood pressure regulation and could be implicated with the manifestation of hypertension. Furthermore, since many of the increased metabolites in the kidneys do not appear proportionally elevated in urine, it is assumed that a significant retention of metabolites occurs in kidneys, perhaps due to alteration of intra-renal metabolite composition, dysfunction in their metabolism, or by water retention in the kidneys [43].

The ADMA showed differences in two types of samples (kidney and urine) and was identified as a feature corresponding to ADMA’s mass, and it was also attributed to two features corresponding to products of ADMA’s metabolism. Non-metabolized ADMA was detected in all samples and increased in kidneys of the Cfz group, whereas the respective levels of ADMA in plasma and urine Cfz are slightly lower than in the Control. This fact implied that, in addition to ADMA’s retention in kidneys, there is also increase in renal biosynthesis in mice treated with Cfz. One metabolic product of ADMA is also increased in the kidneys of treated mice. The compound was excreted as ADMA and ADMA + SO_3_, with the latter being detected only in the urine of treated mice, suggesting that the de novo metabolic pathways are triggered in the kidneys under the impact of Cfz The distribution of ADMA in the bio-samples is shown in Figure 5. The total of all ADMA forms in plasma, the kidneys, and urine appears to be 2.4-fold lower in Cfz-treated vs. the Control mice.

### 3.7. Discovery of New Potential Biomarkers of Cfz-Related Nephrotoxicity

The current study achieved the goal of determining and identifying more than 40 compounds (metabolites and products of metabolites’ metabolism) of proven diagnostic ability (AUC value = 1), that could be potential biomarkers of Cfz-related nephrotoxicity, as follows: kidney (25), urine (16), and plasma (1). So far, only two blood biomarkers (creatinine and urea) are used for the clinical diagnosis of Cfz cardiorenal toxicity [1,44]. Thus, the discovery of this number of potential biomarkers may have a great impact in the prediction of Cfz’s renal adverse effects and, therefore, those compounds should be verified in clinical samples.

## 4. Materials and Methods

### 4.1. Sample Collection and Storage

Τhis study employed plasma, kidney, and urine samples of 12 male C57Bl/6J (13–14 weeks of age) mice. The laboratory animals were bred and housed in the Animal Facility of the Biomedical Research Foundation, Academy of Athens. All in vivo experiments were carried out in accordance with the “Guide for the care and use of Laboratory animals” and experiments were approved by the Ethics Committee (Approval No: 182464;14-05-2019). The mice were housed and maintained according to the ARRIVE guidelines [45]. The animals were randomized in two groups (n = 6 for each group) as follows: (i) Control (NaCl 0.9%), (ii) Cfz (8 mg/kg) for 6 days [1]. The NaCl and Cfz were injected intraperitoneally on alternate days, and at the end of the experiments mice were euthanized by a high dose of ketamine (100 mg/kg) and subsequent cervical dislocation. Mice were placed in metabolic cages for 24 h for urine collection, and they were provided with food and water ad libitum. Plasma samples were collected by centrifugation of heparinized whole blood at 5000 RPM for 15 min. The bio-samples (plasma, kidneys, urine) were collected at the end of the experiments and stored at −80 °C. Carfilzomib regimens were based on our previous study addressing its cardiotoxicity and are translationally equivalent to human doses [3]. Briefly, in humans, Cfz initial dosing is selected to be 27 or 56 mg/m^2^ and can be reduced to 15 mg/m^2^ upon manifestation of life-threatening cardiorenal adverse events, before discontinuation of the therapy. In a translational scope, the dose regimen selected for the four-dose protocol is equivalent to a HED of 29.65 mg/m^2^, which is within the range of the initiation dose of Carfilzomib.

### 4.2. Reagents and Solutions

All the reagents used were of high purity. Methanol and acetonitrile (LC–MS grade) were purchased from Merck (Darmstadt, Germany), ammonium formate was from Fischer Scientific (Geel, Belgium), and formic acid was from Sigma-Aldrich (Steinheim, Germany). Yohimbine hydrochloride and reserpine pharmaceutical grade and primary standards were used as internal standards and were purchased from Merck (Darmstadt, Germany). Distilled water was produced by a Milli-Q purification apparatus (Millipore Direct-Q UV, Bedford, MA, USA).

The preparation of the mobile phase was as follows. For the positive ionization mode, the mobile phase A-pos was an aqueous solution of 5 mM ammonium formate, acidified with 0.01% formic acid, while the mobile phase B-pos was a buffer consisting of acetonitrile–water (95:5 *v*/*v*) containing 5 mM ammonium formate and acidified with 0.01% formic acid. For the negative mode, the mobile phase A-neg was an aqueous solution of 10 mM ammonium formate, and the mobile phase B-neg was a buffer consisting of acetonitrile–water (95:5 *v*/*v*) containing 10 mM ammonium formate.

The preparation of the internal standard (IS) mix solution was as follows. Two stock solutions for yohimbine hydrochloride and reserpine were prepared using ultra-pure water, with a final concentration of 10 mg/L. Both stock solutions were used to prepare the final mixed IS standard solution, consisting of acetonitrile–water (95:5 *v*/*v*) with 1 mg/L final concentration of yohimbine and reserpine.

For the instrument calibration, a calibrant solution of sodium formate dissolved in 2-propanol water (1:1 *v*/*v*) was employed.

### 4.3. Sample Preparation

Samples from three bio-samples, i.e., plasma, kidney, and urine were employed. Different experimental protocols were implemented for the extraction of metabolites from each sample type. In order to avoid the discrimination of some metabolite classes, the sample pre-treatment protocol involved only a protein precipitation step.

The plasma extraction procedure was as follows: 600 μL of frozen methanol was added to 200 μL of sample and mixed by vortexing for 20 s, before being centrifuged using a NEYA 16R centrifugation apparatus (REMI, Mumbai, India) at 10,000× *g* rpm, 5 min, 4 °C. A 350 μL aliquot of the supernatant was evaporated to dryness by a HyperVAC-LITE centrifugal vacuum concentration (Hanil Scientific Inc., Gimpo, Korea). Samples were stored at −80 °C, and reconstituted before the analysis with 150 μL of IS mix solution [16,18,46,47].

The urine extraction procedure was as follows: 500 μL of the sample was centrifuged (10,000× *g* rpm, 5 min, 4 °C) to precipitate particles. The supernatant was diluted with 1000 μL of a methanol–water solution (1:1 *v*/*v*) and an aliquot of 600 μL was evaporated to dryness, stored at −80 °C, and reconstituted with 150 μL of IS mix solution [18,48,49,50]. The acquired data were corrected using the total volume of excreted urine of each mouse.

The kidney extraction procedure was as follows. Kidneys were weighted and mixed with an appropriate volume of a methanol–water solution (1:1 *v*/*v*), adjusted to the sample’s weight; for every 100 mg of tissue 1000 μL of solution were added. The sample was homogenized using the tissue homogenizing CKMix lysing kit (Bertin Corp., Rockville, MD, USA) and the CRYOLYS EVOLUTION tissue homogenizer (Bertin Instruments, Rockville, MD, USA). Homogenization was accomplished in two rounds; initially the sample tissue with the half of the aforementioned solution was submitted to the “hard” mode (9600× *g* rpm, three 20 s cycles followed by 60 s pause) of the homogenizer, and then the blend was centrifuged at 10,000× *g* rpm for 10 min and the supernatant was placed in a 10 mL falcon. The rest of the solution was added in the homogenizing tube with the tissue remainder and submitted to a second cycle of a “soft” mode (5000× *g* rpm, one 60 s cycle) homogenization. After centrifugation, the supernatant was mixed with the one obtained by the first homogenization cycle and vortexed for 10 s. An aliquot of 500 μL of the total extract was evaporated until dryness, stored at −80 °C, and reconstituted with 150 μL of IS mix solution before the LC–MS analysis.

### 4.4. UPLC-ESI-QTOFMS Analysis

The chromatographic separation was accomplished with an ACQUITY UPLC BEH Amide column, 2.1 × 100 mm, 1.7 μm (Waters, Ireland, Dublin), equipped with an ACQUITY UPLC BEH Amide VanGuard Pre-column, 1.7 µm, 2.1 mm × 5 mm (Waters, Ireland, Dublin). The data were acquired by implementing the Dionex UltiMate 3000 RSLC UHPLC system (Thermo Fischer Scientific, Dreieich, Germany) coupled to a Maxis Impact QTOF mass spectrometer (Bruker Daltonics, Bremen, Germany) through an electrospray ionization source (ESI) capable of both positive and negative ionization. The column temperature was maintained at 30 °C. The gradient elution program is the same in both ionization modes. The conditions of liquid chromatography and the settings of ESI–QTOF instrumentation are described in Table 4**.**

### 4.5. Data Acquisition

Before the beginning of data acquisition, the QTOF system was calibrated by direct infusion of sodium formate solution, for the *m*/*z* 100–900 Da range, using the HPC algorithm. The *m*/*z* width was set at 1 mDa and the calibration was acceptable when the score value was higher than 99% and the standard deviation of *m*/*z* error (ppm) was lower than 0.5. For each ionization polarity, all samples were analyzed in the same batch. Three types of QC samples were used, one for each type of bio-sample, made as pooled samples consisting of equal aliquots of all samples from the bio-sample. For each dataset, three QCs were analyzed at the beginning and at the end of the acquisition, while during the acquisition, three samples’ injections were followed by one QC injection. The injection volume was 5 μL. The data were acquired by employing the broadband collision-induced dissociation (bbCID) mode, which belongs to DIA methodologies. In the bbCID mode, low/high CE–MS data are recorded in alternating scans.

### 4.6. Data Pre-Processing

Six complete studies were performed i.e., three types of bio-samples in positive and negative ionization, namely plasma (+), plasma (−), kidney (+), kidney (−), urine (+), urine (−), thus, six datasets were submitted to further processing.

Data collected from DIA were processed as shown in the Figure 6. Samples and QCs were exported as mzXML files using DataAnalysis (Bruker Daltonics, Bremen, Germany) and imported into MZmine 2.51 [13]. Initially, the information corresponding to high collision energy were cropped and removed from the processing file. The data treatment procedure was applied to the low collision energy spectra, as this piece of information represents the biomarkers in their intact form. The peaks corresponding to isotopes and adducts were excluded from further statistical analysis but retained for the annotation. The MZmine parameters are summarized in Table 5.

The obtained peak lists were submitted to QC-based signal correction using statTarget2 [17]. The QC-RLSC algorithm, based on the locally weighted scatterplot smoothing non-parametric regression (LOWESS), was employed [18]. Non-zero variables with values lower than 70% were removed from the dataset, whereas the default parameters of QCspan and CV% cutoff were used. The efficiency of signal correction was evaluated by PCA, with all QCs being included to a tight cluster, as shown in Figure 7.

### 4.7. Multivariate Analysis

The SIMCA 14.1 software (Umetrics, Sweden) was used for the multivariate analysis. Here, PCA modeling was applied to investigate whether the administration of Cfz affects the metabolic profile of plasma, kidneys, and urine. Subsequently, PLS-DA was employed to point out the discriminant variables (corresponding to mz_t_R_ features) between the Cfz and the Control groups. For both PCA and PLS-DA, unit variance (UV) and Pareto data scaling, combined with different types of data transformation, were tested. The tested transformation algorithms did not provide any improvement in the normality of the data and, therefore, no transformation methodology was employed. The UV scaling afforded better clustering; therefore, it was used in all cases. Permutation testing (100 random permutations) was used to evaluate their validity of the PLS-DA models and estimate the degree of overfitting.

### 4.8. Univariate Analysis

MetaboAnalyst 5.0 was used for the univariate analysis which included ROC (receiver operating characteristic) curves, FDR-corrected t-tests, fold change analysis, and volcano plots with fold change threshold [19]. The steps of both multivariate and univariate analysis are shown in Figure 7.

### 4.9. Peaks Identification Procedure

The most significant variables i.e., those with VIP values higher than 1.5 and AUC values higher than 0.9, were selected for identification. DataAnalysis (Bruker Daltonics, Bremen, Germany) was employed to extract ion chromatograms with a specific *m*/*z* (tolerance: 10 ppm) and RT (tolerance: 0.2 min) of the selected features and to “clean” their spectra from background noise. The “cleaned” low and high CE spectra of each selected feature were imported into the RamClustR [51] R-based package in order to assign a pseudo-MSMS to their corresponding precursor ions. Each low CE ion combined with its pseudo-MSMS was searched online and confirmed by the comparison to the reference metabolite MS2 spectra, or its in-silico fragmentation, available in MCID and in the HMDB 5.0 database [20].

The myCompoundID online library was used for peak identification, as the library includes information of both human metabolites and of their metabolism-products [52]. All types of precursor ion adducts and both *no reaction* and *1 reaction* mode were investigated. For the experimental pseudo-MSMS spectra, we used the predicted MS2 spectra as provided by the MyCompound ID library, which uses the HMDB and the Evidence-based Metabolome Library (EML). In the cases of the non-metabolized metabolites, the MS2 was confirmed by comparison to the experimental spectra existing in HMDB. As universal requirements, *m*/*z* tolerance was set to 5 mDa for the precursor ions and to 10 mDa for the fragments. The results were evaluated with their initial score (>0.95) corresponding to formula prediction and with their fit score (>0.75), corresponding to a similarity between experimental and reference MS2 spectra. In the cases of tie between two potential metabolites, their biological disposition was considered. The identification procedure is described in Figure 8.

### 4.10. Data-Driven Suspect Screening of Metabolites

Inspired from the outcome of the above procedures and attempting to investigate the correlations of metabolites among the plasma, kidneys, and urinary system, a hypothesis driven metabolites determination was designed. One list of “suspect-compounds” was created for each polarity and applied to all samples, regardless the bio-sample. These “suspect-lists” included all the identified metabolites and their possible metabolism products. The TASQ Client 2.1 software (Bruker Daltonics, Bremen, Germany) was used to perform the target screening analysis and the semi-quantitation thereof by peak integration. Each sample was subjected to software-based internal calibration, employing a sodium formate spectrum for the calibration. For the compound detection, mass tolerance was set to 5 mDa, RT tolerance to 0.5 min, and the signal to noise (S/N) level was set at 10.

### 4.11. Exploration of Metabolites Alterations between Different Bio-Samples

The above results were used to plot diagrams with the mean and standard deviation (SD) of the detected metabolites’ signal. The mean and the SD were estimated for each group and bio-sample type, aiming to highlight correlations and patterns among the blood, the renal, and the urinary system due to the Cfz administration.

## 5. Conclusions

Carfilzomib is an authorized anti-cancer drug for the treatment of relapsed/refractory multiple myeloma. The drug has been associated with cardiorenal adverse events of unknown pathobiology. Thus, a high-throughput untargeted metabolomics study was implemented to enlighten aspects of nephrotoxicity effects due to Cfz administration. For the metabolomics study, plasma, kidney, and urine samples obtained from treated and Control mice were employed, creating six sets of data (two ionization polarities applied on three bio-sample types).

The results show that kidney and urine were mostly affected, whereas plasma maintained its homeostasis after the drug administration. This fact was confirmed from the statistical analysis, whereas metabolites found up or down regulated were in accordance with an acute renal dysfunction. This was also verified from by the inter-organ correlation. As a general observation, kidney and urine samples of Cfz treated mice show higher numbers of differentially regulated metabolites, indicating a potential increase in the renal metabolism. The inter-organ correlation study pointed out several metabolites, increased only in kidney but not in urine and plasma samples. Furthermore, several metabolites were detected only in the urine samples. Thus, it is assumed that Cfz (i) elevates renal metabolic rate, which may induce high consumption of oxygen, and (ii) provokes retention of metabolites, although this is not expressed in the plasma levels of these compounds.

The identification procedure ended up with 67% of identified features. The identified biomarkers revealed a potential explanation of both Cfz’s renal toxicity and the cause of retention in kidneys, as several of them are already associated with renal failure or kidney injury; however, in the literature, the majority of them referred to blood samples. Thus, it is assumed that Cfz influences renal metabolome in several directions, dysregulating more than one biological pathways, and causing locally estimated damage to renal function. Three main mechanisms seem to co-operate for this local kidney injury, as follows: (a) inhibition of NO production (via ADMA, 2PY, and PAA) resulting to kidney resistance elevation, (b) increase in oxidative stress (through ac4C and 2-aminoisobutyric acid), and (c) inflammation and kidney injury (due to acC4, PAA, 2PY and 2-aminoisobutyric acid) under the influence of the NO decrease. This local damage to kidneys may be the cause of the component’s retention.

## Figures and Tables

**Figure 1 molecules-27-07929-f001:**
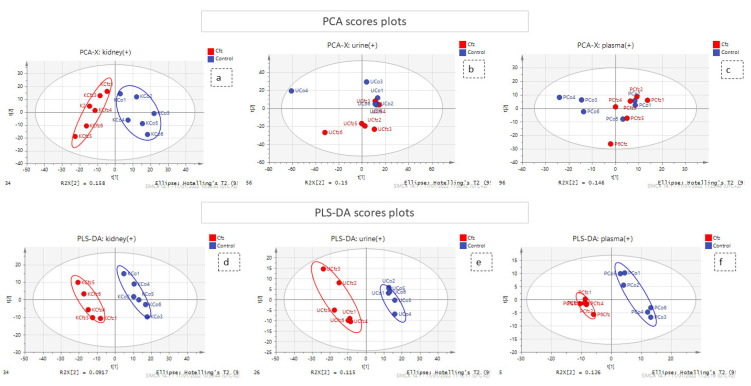
PCA and PLS-DA score plots. The red and the blue points of the score plots represent the Cfz group samples and the Control group samples, respectively. (**a**) PCA of kidney (+) dataset; (**b**) PCA of urine (+) dataset; (**c**) PCA of plasma (+) dataset; (**d**) PLS-DA of kidney (+) dataset; (**e**) PLS-DA of urine (+) dataset; (**f**) PLS-DA of plasma (+) dataset.

**Figure 2 molecules-27-07929-f002:**
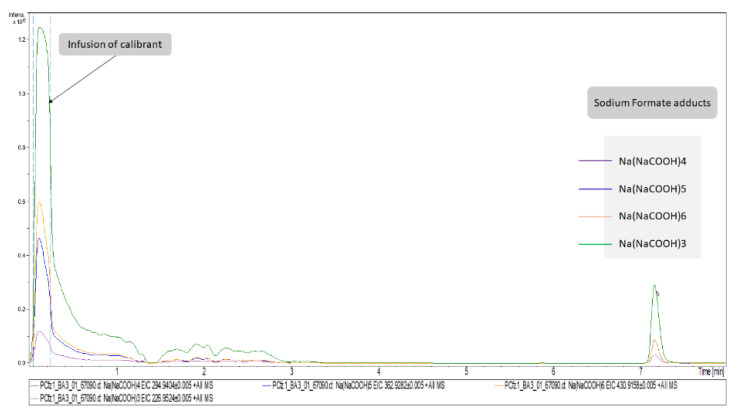
Representation of plasma (+)’s most important variables as extracted ion chromatograms. The first segment of these EICs corresponds to signals obtained from the infusion of calibrant solution. The chromatographic peaks at RT = 7.16 correspond to the adducts of formate detected in Cfz plasma samples.

**Figure 3 molecules-27-07929-f003:**
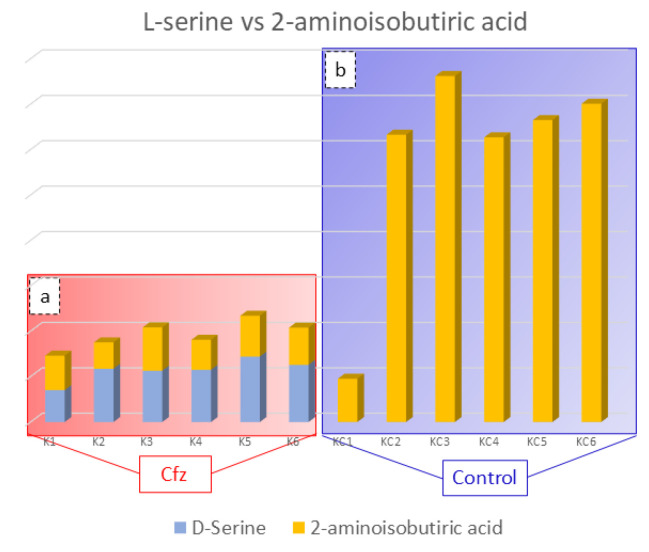
Bar charts representing D-serine (blue color) and 2-aminoisobutyric acid (orange color) content in the Cfz (**a**) and Control (**b**) kidney samples. Aminoisobutyric acid is decreased in Cfz samples and D-serine is increased, while the later was not detected in Control samples. This suggested that in the Cfz case, the 2-Aminoisobutyric acid is decreased and is not able to inhibit D-Serine from H_2_O_2_ production and, therefore, cannot protect the kidneys from oxidative stress.

**Figure 4 molecules-27-07929-f004:**
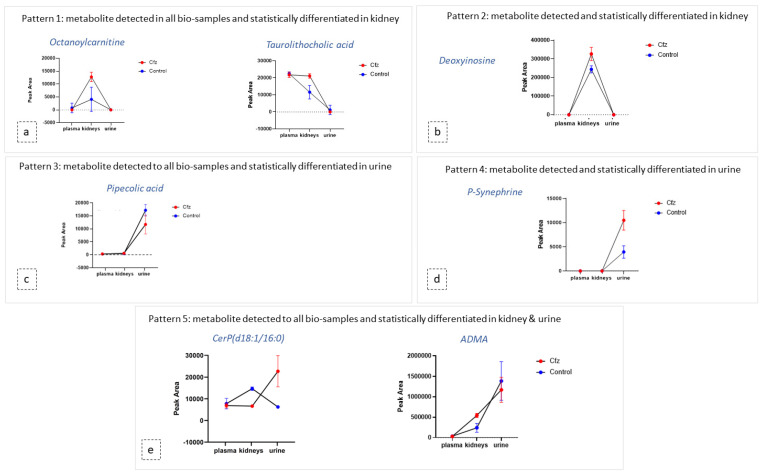
Diagrams of metabolite content (expressed via mean and SD) in the plasma, kidneys, and urine of both Cfz (red) and Control (blue) groups. Each diagram represents a type of the revealed patterns of metabolite distribution among bio-samples: (**a**) metabolites detected in all biosamples and statistically differentiated in kidney; (**b**) metabolites detected and statistically differentiated in kidney; (**c**) metabolites detected in all biosamples and statistically differentiated in urine; (**d**) metabolites detected and statistically differentiated in kidney (**e**) metabolites detected in all biosamples and statistically differentiated in kidney and urine.

**Figure 5 molecules-27-07929-f005:**
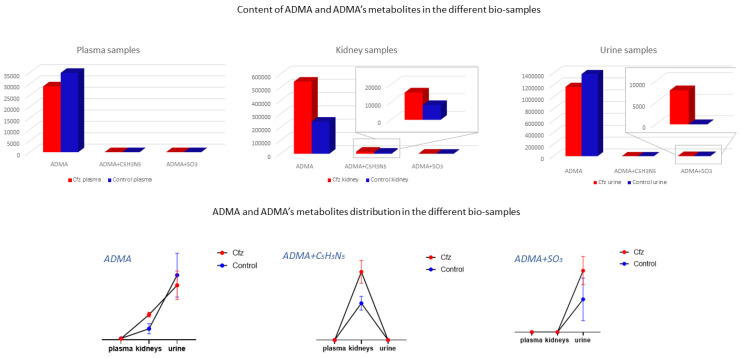
ADMA had been detected as an intact metabolite and also in two forms of its metabolism. Here, ADMA, per se, is increased in Cfz kidneys; however, it is decreased in Cfz plasma and urine, suggesting an increased rate of ADMA production or ADMA’s strong retention at the kidney level. Furthermore, ADMA’s metabolites (ADMA + C_5_H_3_N_5_ and ADMA + SO_3_) have been highly detected in Cfz kidneys and urine.

**Figure 6 molecules-27-07929-f006:**
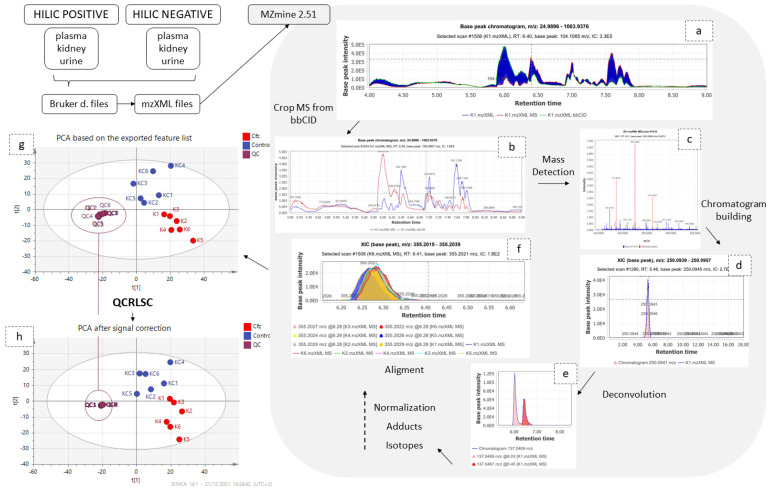
Graphical description of data pre-processing workflow. The low and high CE–MS before (**a**) and after (**b**) their separation; (**c**) set of noise levels for the mass detection; (**d**) one peak which resulted from the chromatogram building; (**e**) the chromatogram deconvolution; (**f**) the result of the alignment; the PCAs before (**g**) and after (**h**) signal correction.

**Figure 7 molecules-27-07929-f007:**
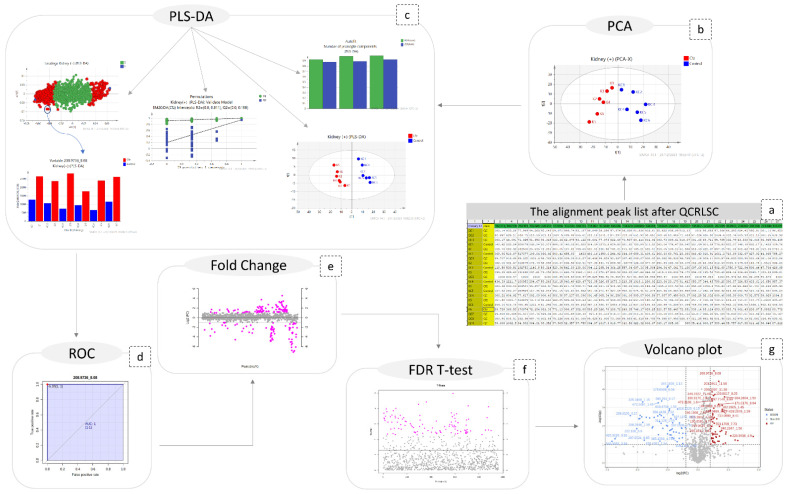
Graphical description of the statistical workflow for kidney (+), as follows: (**a**) the final peak list; (**b**) PCA; (**c**) PLS-DA; ( the number of PCs; the scores-plot the loadings plot; the content plot of a discriminant variable; the result of permutations test); (**d**) the ROC curve of a biomarker; (**e**) the fold change plot; (**f**) FDR-TT representation; (**g**) volcano plot.

**Figure 8 molecules-27-07929-f008:**
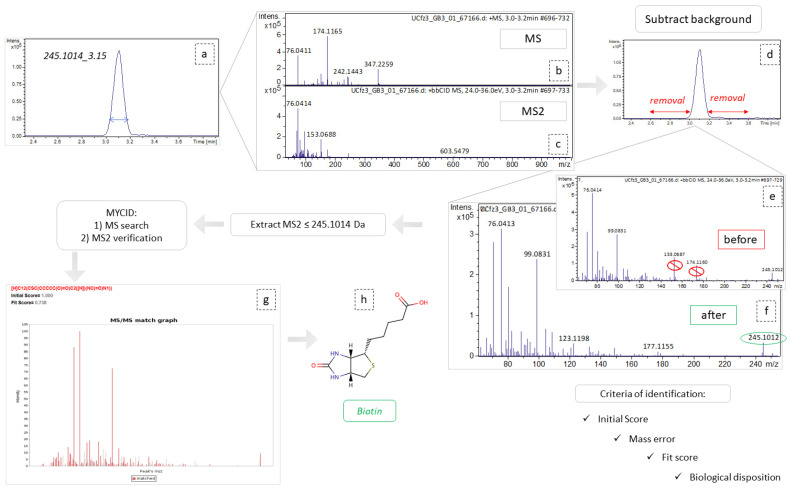
Graphical representation of identification workflow, employing the example of variable 254.1014_3.15. (**a**) Extraction of ion chromatogram, *m*/*z* = 245.1014, t_R_ = 3.14; (**b**,**c**) refer to MS and MS2 spectra, respectively, corresponding to this peak area; (**d**) the area removed as background spectra; (**e**,**f**) represent the MS2 spectra before and after background subtraction, respectively; (**g**) MSMS match graph obtained from MCID, during the identification procedure; (**h**) the structural representation of *biotin*.

**Table 1 molecules-27-07929-t001:** Summary of the features which resulted from the peak-picking (Detected), those retained from QC-RLSC-based signal correction, performed via statTarget2 (*After QC-RLCS*), and those included for multivariate analysis that had values different from their median (*Used for multivariate analysis*).

	Number of Variables:		
Dataset	Detected	After QC-RLCS	Used for Multivariate Analysis
Plasma (+)	624	346	191
Plasma (−)	156	67	49
Kidney (+)	1079	964	964
Kidney (−)	239	195	195
Urine (+)	1769	1509	684
Urine (−)	533	458	328

**Table 2 molecules-27-07929-t002:** Summary of results from the multivariate and univariate statistical analysis. The first six columns are referring to PLS-DA classification, namely the number of principle components (PCs), measure of fit (R2), prediction ability (Q2), the higher estimated VIP values (*Higher VIP value*), and the results of the permutations test ((Q2) and (R2)). The last four columns show the number of variables that exceed the set limit, i.e., 11 variables of the plasma (+) dataset had a VIP value higher than 1.5.

							Number of Biomarkers			
	PLS-DA Model				PLS-DA Permutations Test		PLS-DA Analysis	ROC Analysis	FDR-*t*-Test	Fold-Change
	PCs	Q2	R2	Higher VIP Value	(Q2)	(R2)	VIP > 1.5	AUC > 0.9	*p*-Value < 0.05	log2(FC) > 2
Plasma (+)	3	0.5	0.99	2.3	0.32	0.99	11	8	20	107
Plasma (−)	4	0.14	0.92	2.53	0.27	0.94	13	0	0	24
Kidney (+)	3	0.79	0.99	1.96	0.13	0.98	40	105	110	79
Kidney (−)	3	0.93	0.99	1.73	0.38	0.97	21	45	30	29
Urine (+)	4	0.86	1	2.35	0.58	0.99	23	68	82	926
Urine (−)	3	0.77	0.99	2.49	0.20	0.99	19	41	40	158

**Table 3 molecules-27-07929-t003:** Identification table. The identification procedure was performed to those features with VIP and AUC values of >1.5 and >0.9, respectively. The MyCompoundID (MCID) online library was used, and both *no-metabolic reaction* and *one-metabolic reaction* have been considered for the identification. Initial and fit score are related to identification efficiency. *Initial score* evaluates the relativity between the theoretical formula and the experimental *m*/*z*, and the *fit score* evaluates the matching between the reference and the experimental or in-silico MSMS spectra. * No reaction refers to metabolites which have not undergone any metabolic reaction, while the entry to the column is attributed to the metabolic reaction i.e., the addition or loss of a corresponding group, ** The (↑) arrow denotes increased levels of the metabolites in Cfz-samples compared to the control and; (↓) arrow denotes the decreased levels of the metabolites in Cfz-samples compared to the control.

Precursor Mass Exp.	Sample Type	ESI Polarity	RT	Cfz-Regulation **	VIP	AUC	Formula	Compound	HMDB	MCID Initial Score	MCID Fit Score	Reactions of Metabolism *	Precursor Type	Precursor Mass Theo.	Error (mDa)
**153.0695**	urine	+	3.14	↑	1.64	1	C_7_H_8_N_2_O_2_	N1-Methyl-2-pyridone-5-carboxamide	HMDB0004193	1	0.87	NO REACTION	M + H	153.0659	−0.004
**551.2665**	urine	+	4.51	↑	1.62	1	UNKNOWN	-							
**135.0948**	urine	+	6.1	↑	1.47	1	C_3_H_7_N_3_O_2_	Guanidoacetic acid	HMDB0001528	1	0.52	[+NH_3_]	M + H	135.0877	−0.007
**286.1103**	urine	+	5.78	↑	1.44	1	C_11_H_15_N_3_O_6_	N4-Acetylcytidine	HMDB0005923	1	0.72	NO REACTION	M + H	286.1034	0.007
**183.1168**	urine	+	3.28	↑	1.40	1	C_9_H_13_NO_2_	p-Synephrine	HMDB0004826	0.96	0.59	[+NH]	M + H	183.1128	−0.004
**245.1014**	urine	+	3.11	↑	1.40	1	C_10_H_16_N2O_3_S	Biotin	HMDB0000030	1	0.74	NO REACTION	M + H	245.0954	−0.006
**181.1626**	urine	+	1.35	↑	1.39	1	C_10_H_16_O	Perillyl alcohol	HMDB0003634	0.99	0.68	[+C_2_H_4_]	M + H	181.1587	−0.004
**214.1852**	urine	+	1.36	↑	1.38	1	C_13_H_23_NO_4_	2-Hexenoylcarnitine	HMDB0013161	1	0.92	[−CO_2_]	M + H	214.1802	−0.004
**199.1739**	urine	+	1.35	↑	1.35	1	C_10_H_20_O	Decanal	HMDB0011623	0.86	0.83	[+C_2_H_2_O]	M + H	199.1693	−0.005
**283.1111**	urine	+	5.13	↑	1.32	0.96	C_8_H_18_N_4_O_2_	Asymmetric dimethylarginine	HMDB0001539	0.96	0.55	[+SO_3_]	M + H	283.1037	−0.007
**112.1154**	urine	+	6	↑	1.40	0.96	UNKNOWN	-							
**392.2368**	urine	+	5.4	↑	1.34	0.9	UNKNOWN	-							
**144.9639**	urine	−	1.03	↑	1.48	1	C_6_H_4_Cl_2_O	2,4-Dichlorophenol	HMDB0004811	1	0.79	[−O]	M-H	144.9606	−0.003
**305.1473**	urine	−	6.07	↑	1.40	1	C_14_H_18_N_2_O_4_	Phenylalanyl-hydroxyproline	HMDB0011176	1	0.844	[+C_2_H_4_]	M-H	305.1496	0.002
**279.0149**	urine	−	1.14	↑	1.33	1	UNKNOWN	-							
**258.9891**	urine	−	4.22	↑			C_9_H_8_O_4_	4-Hydroxyphenylpyruvic acid	HMDB0000707	0.99	0.831	[+HPO_3_]	M-H	259.0002	0.011
**215.0002**	urine	−	1.17	↑	1.29	1	C_8_H_8_O_2_	Phenylacetic acid	HMDB0000209	1	0.77	[+HPO_3_]	M-H	215.0104	0.010
**363.0135**	urine	−	1.55	↑	1.28	1	C_3_H_6_O_3_S	3-Mercaptolactic acid	HMDB0002127	0.81	0.78	[+C_6_H_11_O_8_P]	M-H	363.0145	0.001
**123.0116**	urine	−	5.27	↑	1.23	0.9	UNKNOWN	-							
**324.9654**	urine	−	4.75	↑	1.23	0.96	UNKNOWN	-							
**365.0294**	urine	−	1.12	↑	1.21	0.9	C_10_H_15_N_2_O_9_P	Imidazoleacetic acid-ribotide	HMDB0006032	0.86	0.75	[+CO]	M-H	365.0381	0.009
**199.9947**	urine	−	2.1	↓	1.30	1	C_3_H_8_NO_6_P	Phosphoserine	HMDB0000272	1	0.75	[+O]	M-H	199.9955	0.001
**230.9946**	urine	−	1.37	↓	1.30	1	C_8_H_8_O_3_	4-Hydroxy-3-methylbenzoic acid	HMDB0004815	1	0.9	[+SO_3_]	M-H	230.9958	0.001
**144.0655**	urine	−	2.27	↓	1.20	1	C_6_H_11_NO_2_	Pipecolic acid	HMDB0000070	1	0.76	[+O]	M-H	144.0655	0.000
**337.0345**	urine	−	1.63	↓	1.29	1	C_5_H_11_O_8_P	D-Arabinose 5-phosphate	HMDB0011734	0.72	0.8	[ +C_5_H_4_N_2_O]	M-H	337.0431	0.009
**208.9736**	kidney	+	8.08	↑	1.38	1	UNKNOWN	-							
**349.2322**	kidney	+	11.68	↑	1.35	1	C_15_H_29_NO_4_	Octanoylcarnitine	HMDB0000791	1	0.71	[+CO2]	M + NH4	349.2333	0.001
**336.1931**	kidney	+	6.97	↑	1.31	1	C_8_H_18_N_4_O_2_	Asymmetric dimethylarginine	HMDB0001539	1	0.46	[+C_5_H_3_N_5_]	M + H	336.1891	−0.004
**133.0617**	kidney	+	8.06	↑	1.31	1	C_3_H_8_N_2_O_2_	2,3-Diaminopropionic acid	HMDB0002006	0.9	0.52	[+CO]	M + H	133.0608	−0.001
**245.0777**	kidney	+	5.82	↑	1.30	1	C_9_H_12_N_2_O_6_	Uridine	HMDB0000296	1	0.4	NO REACTION	M + H	245.0768	−0.001
**160.5245**	kidney	+	7.82	↑	1.29	1	UNKNOWN	-							
**384.2604**	kidney	+	1.59	↑	1.28	1	UNKNOWN	-							
**203.1507**	kidney	+	11.58	↑	1.28	1	C_8_H_18_N_4_O_2_	Asymmetric dimethylarginine	HMDB0001539	1	0.76	NO REACTION	M + H	203.1503	0.000
**297.7149**	kidney	+	6.99	↑	1.25	1	C_10_H_20_O_3_	3-Hydroxycapric acid	HMDB0002203	1	0.58	[+C_5_H_4_N_2_O]	M + H	297.1809	−0.534
**258.4705**	kidney	+	7.72	↑	1.24	1	UNKNOWN	-							
**166.0867**	kidney	+	6.97	↑	1.24	1	C_9_H_10_O_2_	4-Ethylbenzoic acid	HMDB0002097	1	0.63	[+NH]	M + H	166.0863	0.000
**120.42**	kidney	+	6.97	↑	1.24	0.97	C_8_H_11_N	1-Phenylethylamine	HMDB0002017	0.96	0.71	[−H2]	M + H	120.0808	−0.339
**331.1662**	kidney	+	6.96	↑	1.23	1	C_18_H_21_NO_4_	(S)-3-Hydroxy-N-methylcoclaurine	HMDB0006921	1	0.7	[+NH]	M + H	331.1652	−0.001
**166.8851**	kidney	+	6.95	↑	1.23	1	UNKNOWN	-							
**171.0176**	kidney	+	8.04	↑	1.22	1	C_4_H_8_N_2_O_3_	Ureidopropionic acid	HMDB0000026	0.77	0.61	NO REACTION	M + K	171.0167	−0.001
**600.4706**	kidney	+	1.53	↓	1.25	1	C_34_H_68_NO_6_P	CerP(d18:1/16:0)	HMDB0010700	1	0.83	[−H2O]	M + H	600.4751	0.005
**556.4439**	kidney	+	1.52	↓	1.25	1	UNKNOWN	-							
**288.291**	kidney	+	6.17	↓	1.23	1	C_18_H_39_NO_2_	Sphinganine	HMDB0000269	1	0.74	[−CH_2_]	M + H	288.2897	−0.001
**166.4847**	kidney	+	6.96	↑	1.28	0.97	UNKNOWN	-							
**132.1027**	kidney	+	7	↑	1.27	0.97	C_6_H_12_O_2_	L-alpha-Aminobutyric acid	HMDB0000452	0.98	0.82	[+NH]	M + H	132.1019	−0.001
**609.2826**	kidney	+	6.15	↑	1.26	0.97	C_26_H_45_NO_8_S_2_	Taurolithocholic acid 3-sulfate	HMDB0002580	1	0.59	[+CO]	M + NH4	609.2874	0.005
**263.1976**	kidney	+	6.94	↑	1.26	0.97	C_12_H_23_NO_4_	Valerylcarnitine	HMDB0013128	1	0.63	[+NH_3_]	M + H	263.1965	−0.001
**86.3838**	kidney	+	6.94	↑	1.26	0.97	UNKNOWN	-							
**160.9176**	kidney	+	7.82	↑	1.25	0.97	UNKNOWN	-							
**179.0616**	kidney	−	7.46	↑	1.33	1	C_6_H_6_N_4_O_2_	1-Methylxanthine	HMDB0010738	1	0.85	[+CH_2_]	M-H	179.0564	−0.005
**132.0344**	kidney	−	7.83	↑	1.26	1	C_4_H_4_O_4_	Fumaric acid	HMDB0000134	1	0.74	[+NH_3_]	M-H	132.0291	−0.005
**225.0678**	kidney	−	7.9	↑	1.21	1	C_7_H_16_NO_2_	4-Trimethylammoniobutanoic acid	HMDB0001161	1	0.73	[+SO_3_]	M-H	225.0665	−0.001
**130.0914**	kidney	−	6.86	↑	1.20	1	C_6_H_10_O_2_	delta-Hexanolactone	HMDB0000453	1	1	[+NH_3_]	M-H	130.0863	−0.005
**267.0796**	kidney	−	6.6	↑	1.20	1	C_10_H_12_N_4_O_4_	Deoxyinosine	HMDB0000071	1	0.92	[+O]	M-H	267.0724	−0.007
**124.0114**	kidney	−	7.25	↑	1.20	1	UNKNOWN	-					M-H		
**180.0716**	kidney	−	7.34	↑	1.20	1	C_9_H_8_O_3_	Phenylpyruvic acid	HMDB0000205	1	0.82	[+NH_3_]	M-H	180.0655	−0.006
**289.0737**	kidney	−	6.03	↑	1.19	1	C_6_H_12_O_7_	Galactonic acid	HMDB0000565	0.92	0.74	[+C_4_H_2_N_2_O]	M-H	289.0666	−0.007
**203.0883**	kidney	−	6.78	↑	1.19	1	C_11_H_12_N_2_O_2_	L-Tryptophan	HMDB0000929	1	0.85	NO REACTION	M-H	203.0815	−0.007
**306.0639**	kidney	−	6.03	↑	1.18	1	C_14_H_15_NO_7_	Indoxyl glucuronide	HMDB0010319	1	0.78	[−H_2_]	M-H	306.0608	−0.003
**296.8881**	kidney	−	7.07	↑	1.18	1	UNKNOWN	-					M-H		
**171.0116**	kidney	−	7.82	↑	1.18	0.97	C_7_H_8_O_3_S	p-Cresol sulphate	HMDB0011635	1	0.79	NO REACTION	M-H	171.0110	−0.001
**145.0664**	kidney	−	7.88	↑	1.16	0.97	C_5_H_10_N_2_O_3_	L-Glutamine	HMDB0000641	1	0.6	NO REACTION	M-H	145.0608	−0.006
**303.056**	kidney	−	6.61	↑	1.15	1	C_10_H_12_N_2_O_8_	Orotidine	HMDB0000788	1	0.93	[+O]	M-H	303.0459	−0.010
**128.9636**	kidney	−	7.07	↑	1.14	1	UNKNOWN	-					M-H		
**164.0767**	kidney	−	6.83	↑	1.13	1	C_9_H_13_NO_3_	Normetanephrine	HMDB0000819	0.87	0.83	[−H_2_O]	M-H	164.0706	−0.006
**243.0685**	kidney	−	6.03	↑	1.10	1	C_4_H_8_O_5_	Threonic acid	HMDB0000943	0.96	0.85	[+C_5_H_4_N_2_O]	M-H	243.0612	−0.007
**379.107**	kidney	−	6.07	↑	1.11	1	C_15_H_15_NO_4_	L-Thyronine	HMDB0000667	0.96	0.66	[+C_2_H_5_NO_2_S]	M-H	379.0958	−0.011
**302.1068**	kidney	−	7.68	↑	1.11	0.97	C_10_H_17_N_3_O_6_	N2-gamma-Glutamylglutamine	HMDB0011738	1	0.72	[+CO]	M-H	302.0983	−0.009
**294.9404**	plasma	+	7.16	↑	1.45	1	Na(NaCOOH)_4_	Formate	HMDB0303296				M+	294.9389	−0.001
**362.9282**	plasma	+	7.16	↑	1.45	0.97	Na(NaCOOH)_5_	Formate	HMDB0303297				M+	362.9263	−0.002
**430.9158**	plasma	+	7.16	↑	1.37	0.91	Na(NaCOOH)_6_	Formate	HMDB0303298				M+	430.9138	−0.002
**226.9524**	plasma	+	7.16	↑	1.09	0.91	Na(NaCOOH)_3_	Formate	HMDB0303299				M+	226.9515	−0.001
**332.3335**	plasma	+	3.36	↓	1.13	1	UNKNOWN	-							
**304.3021**	plasma	+	3.44	↓	1.23	0.97	UNKNOWN	-							
**326.3804**	plasma	+	3.17	↓	1.35	0.94	UNKNOWN	-							
**717.0657**	plasma	+	6.46	↑	1.36	0.91	UNKNOWN	-							

**Table 4 molecules-27-07929-t004:** The gradient conditions applied on the UPLC system, and the settings for the ionization source (ESI) for the mass analyzer and for the collision cell.

UPLC Gradient Conditions		
% B-(pos/neg)	Time (min)	Flow (mL/min)
**100**	0	0.2
**100**	2	0.2
**5**	10	0.2
**5**	15	0.2
**100**	15.1	0.2
**100**	22	0.2
**ESI–QTOF Settings**		
**Nebulizer gas**		N_2_
**Nebulizer gas pressure**		2 bar
**Drying gas**		N_2_
**Drying gas flow**		10 L/min
**Drying temperature**		200 °C
**Capillary voltage—positive**		3.5 kV
**Capillary voltage—positive**		2.5 kV
**End plate offset**		0.5 kV
**bbCID collision energy**		24 V–36 V (ramp)
** *m* ** **/*z* scan range**		100–1000 Da

**Table 5 molecules-27-07929-t005:** The algorithms and the parameters applied for the peak-picking.

Step	Algorithm	Parameters	Comments
Mass Detection	Centroid Mass detector	Noise level: 500–1500	adjusted to each dataset
Chromatogram building	ADAP Chromatogram builder	Number of scans: 10	
		Group Intensity threshold: 1000–2500	adjusted to each dataset
		Minimum highest intensity: 1000–4000	adjusted to each dataset
		*m*/*z* tolerance: 5 mDa	
Chromatogram deconvolution	Noise amplitude/AUTO mz centre calculation	Minimum peak height: 1100–4100 abs	adjusted to each dataset
		Peak duration range: 0.1–1 min	
		Amplitude of noise: 100–1000	adjusted to each dataset
Isotopes	Isotopic peaks grouper	*m*/*z* tolerance: 5 mDa	
		Retention time tolerance: 0.1	
		Maximum charge: 2	
Adducts	Adducts search	[M + Na], [M + K], [M + MeOH], [M + HCOOH], [M + ACN], etc.	
		RT tolerance: 0.1 min	
		*m*/*z* tolerance: 10 mDa	
		Max relative-adduct peak height: 100%	
Normalization	Retention time calibration	*m*/*z*: tolerance 5 mDa	
		Retention time tolerance: 0.2 min	
		Minimum standard intensity: 5000–10,000 abs	adjusted to each dataset
Alignment	Join aligner	*m*/*z* tolerance: 10 mDa	
		Retention time tolerance: 0.5 min	
Remove duplicates	Duplicate peak finder/New Average	*m*/*z* tolerance: 20 mDa	
		RT tolerance: 0.8 min	
Gap filling	Peak Finder	Intensity tolerance: 20%	
		*m*/*z* tolerance: 10 mDa	
		Retention time tolerance: 0.8 min	

## Data Availability

This data can be found here: GNPS Public Spectral Library, MassIVE Datasets, MassIVE accession: MSV000089596, link: https://gnps.ucsd.edu/ProteoSAFe/result.jsp?task=12e7963068194c9e91c55fcd0b429676&view=advanced_view (accessed on 17 August 2022).

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
