# Peer review of "An Untargeted Metabolomics Approach on Carfilzomib-Induced Nephrotoxicity"

_molecules, 2022, doi:10.3390/molecules27227929_

Round 1

Reviewer 1 Report

The author use untargeted metabolomics profiling to discover potential biomarkers for possible early detecting of Cfz induced nephrotoxicity. The results indicated Cfz might cause several renal dysfunction related metabolites change.

Overall, the experiment design is clear and the conclusion is supported by the results. Please address following questions:

1. The author mentioned they use data independent acquisition (DIA) method, what is the isolation window used? Since DIA usually use a wide isolation window, which means multiple precursor ions will be fragmentated at the same time to generate MS2 spectra, how you preform identification with these complicated MS2 spectra?

2. For identification, do you have your in-house library? Are you use RT as one criteria for identification or you only use MS2 information? As for MS2, do you compare to experimental MS2 library  or in-silico MS2 database?

3. For "Results" part, 2.1 and 2.2 have same title, either merge them together or use different title.

4. For Figure 3, heatmap probably will be better to illustrate your data.

5. For Figure 4, it might be good to use y axis showing the degree of change for each compound from different sample types.

6. For section 4.3, technically, plasma and urine are not tissue, please use the right turn to descript.

7. For Figure 5, what is metabolite 1 and 2, please clarify.

8. Could author clarify the unit: mDa? Probably due to the instrument difference, the MS tolerance reported is usually <10ppm.

Author Response

We would like to thank the reviewers for their constructive comments that gave us the chance to improve our manuscript. We answered to all the issues paused and we believe that we efficiently clarified all the parts that needed improvement. Please find bellow our responses, in plain text, under the reviewers comments that are highlighted with bold. The changes made are reflected at the revised manuscript uses track changes.

The author use untargeted metabolomics profiling to discover potential biomarkers for possible early detecting of Cfz induced nephrotoxicity. The results indicated Cfz might cause several renal dysfunction related metabolites change. Overall, the experiment design is clear and the conclusion is supported by the results. Please address following questions:

The author mentioned they use data independent acquisition (DIA) method, what is the isolation window used? Since DIA usually use a wide isolation window, which means multiple precursor ions will be fragmentated at the same time to generate MS2 spectra, how you preform identification with these complicated MS2 spectra? We thank the reviewer for the chance to clarify this issue. We have used bbCID algorithm as implemented to the Maxis-Impact-QTOFMS instrument. The specific model does not provide the ability of SWATH acquisition i.e., to use narrow windows for the isolation of precursor ions.  Therefore, the acquisition windows of low and high collision energy spectra were the same (50-1000 Dalton). For the identification procedure we employed the RamClustR R-based package which gives the ability of assigning pseudo-MSMS spectra to a precursor ion. This information is now stated in the revised manuscript.

For identification, do you have your in-house library? Are you use RT as one criteria for identification or you only use MS2 information? As for MS2, do you compared to experimental MS2 library or in-silico MS2 database? This is a very interesting question. In this study, we did not use retention time information as additional criteria for the identification, as many of our candidate structures were assigned to metabolic products of metabolites, with no standards available. For the experimental pseudo-MSMS spectra, we used the predicted MS2 spectra as provided by MyCompound ID library, which uses the HMDB and the Evidence-based Metabolome Library (EML). In the cases of the non-metabolized metabolites the MS2 was confirmed by comparison to the experimental spectra existing in HMDB. This is clearly indicated in the revised manuscript.

"Results" part, 2.1 and 2.2 have same title, either merge them together or use different title. Thank you, the typo has been corrected in the revised manuscript.

For Figure 3, heatmap probably will be better to illustrate your data. Thank you very much for your kind suggestion. In our opinion, the bar-charts in Figure 3 are more comprehensible, and describe the decrease of 2-aminoisobutiric acid and the increase of d-serine in Cfz kidney samples, equally to the heatmap. However, the heatmap suggested by the reviewer seemed like a good idea, therefore we incorporated that to the revised supplementary material.

For Figure 4, it might be good to use y axis showing the degree of change for each compound from different sample types. We thank again the reviewer for paying attention in these kinds of details and suggesting revisions that will improve the final format of our text. Thus, in Figure 4 we have included a y axis showing the peak area (reflecting the content) of each metabolite in the samples.

For section 4.3, technically, plasma and urine are not tissue, please use the right turn to descript. Thank you for carefully reading our manuscript. We have chosen the “biosamples” as a more proper description to the revised version.

For Figure 5, what is metabolite 1 and 2, please clarify. Thank you again for pointing this out, metabolite 1 referes to a metabolite  ADMA with the addition of ~C5H3N5 group which corresponds to Guanine and metabolite 2 correspond to a metabolite of ADMA with the addition of ~SO3 group. This information is now included in the revised text.

Could author clarify the unit: mDa? Probably due to the instrument difference, the MS tolerance reported is usually <10ppm. We would like to thank you for giving us the chance to clarify this issue. The used unit for denoting the mass accuracy is mili-Dalton with the symbol mDa. In our laboratory we prefer this unit over ppm as it is independent of the mass range.

Reviewer 2 Report

Comments to Authors:

The topic of the study of this in vivo untargeted metabolomics approach is of major interest as nothing is yet available about the change in the metabolome of different bio-samples. This manuscript includes very intriguing data, which provide novel insights into molecular mechanisms underlying nephrotoxic adverse effects of carfilzomib. The experiments are well designed and the text is well written, so, there are only a few minor considerations that need to be corrected.

Line 57: the authors report that there are several cases of renal failure. This statement must be supported with references

Throughout the manuscript authors often write un-targeted. This term should be replaced by untargeted (example: line 75, line 85, line 617)

Line 129: The title of topic 2.2 is the same as topic 2.1. I think it would make sense for topic 2.2 to be called statistical analysis.

Line 143-144: The authors report that urine separation is more apparent in the negative mode. Visually it may be true, but the values in table 2 state the opposite.

It's not clear to me why authors remove values that are different from the median value. Couldn't these variables be discriminating/informative?

Line 162-163: The authors report that “PLS-DA analysis succeeded classification between the Cfz and the control group for all datasets”. Once again I say that visually it may be true, but the values in table 2 indicate that for the negative mode plasma this is not at all true, since the value of Q2 is extremely low, and the value of Q2 in the permutation test turned out to be higher.

Line 433: the authors report that samples (plasma, kidney and urine) were collected during and at the end of the experiments. I didn't realize at what time-point each of the samples was collected. I thought they were all collected at the end of the experiment. Can you please clarify.

Sometimes authors refer to the three samples as tissues (example line 463 and 525), which is incorrect as only the kidney is a tissue. Urine and plasma are biofluids. So it will be more correct to treat the 3 by biological samples.

Line 484 and line 487: can you please clarify the difference between “hard” and “soft” homogenization? What are the experimental conditions for carrying out each of them?

I think it would be extremely important to include the PCA with the projection of QCs for all datasets as a supplementary material.

Line 548: The authors refer that they tested different types of transformations. But it was not clear whether the final data was transformed or not? If yes, what method was applied?

Line 548-549: Different scaling methods emphasizes and suppresses different kinds of features. Please specify and discuss how UV scaling affected your data. 

Table 5: How were the pre-processing parameter determined or optimized?

Author Response

We would like to thank the reviewers for their constructive comments that gave us the chance to improve our manuscript. We answered to all the issues paused and we believe that we efficiently clarified all the parts that needed improvement. Please find bellow our responses, in plain text, under the reviewers comments that are highlighted with bold. The changes made are reflected at the revised

The topic of the study of this in vivo untargeted metabolomics approach is of major interest as nothing is yet available about the change in the metabolome of different bio-samples. This manuscript includes very intriguing data, which provide novel insights into molecular mechanisms underlying nephrotoxic adverse effects of carfilzomib. The experiments are well designed and the text is well written, so, there are only a few minor considerations that need to be corrected.

Line 57: the authors report that there are several cases of renal failure. This statement must be supported with references. We thank the reviewer for pointing out this issue. We have clarify the corresponding references in the revised manuscript.

Throughout the manuscript authors often write un-targeted. This term should be replaced by untargeted (example: line 75, line 85, line 617). We thank you a lot for carefully reviewing our manuscript. Your observation is correct, and we have replaced this term in our revised text.

Line 129: The title of topic 2.2 is the same as topic 2.1. I think it would make sense for topic 2.2 to be called statistical analysis. We have corrected this typo, thank you a lot for pointing this out.

Line 143-144: The authors report that urine separation is more apparent in the negative mode. Visually it may be true, but the values in table 2 state the opposite.   

It's not clear to me why authors remove values that are different from the median value. Couldn't these variables be discriminating/informative? We thank the reviewer for this comment. We would like to clarify that the “separation tendency” described by the figure 1 (supplementary material) refers to the PCA model of urine (-) while the table 2 refers to the corresponding PLS-DA models of the same data. Concerning the removal of values non-differing from the median value; is a frequent step in data treatment as it eliminates values with low informative content. Thus, we shall clarify that we only removed such values, as it is described in the manuscript.

Line 162-163: The authors report that “PLS-DA analysis succeeded classification between the Cfz and the control group for all datasets”. Once again I say that visually it may be true, but the values in table 2 indicate that for the negative mode plasma this is not at all true, since the value of Q2 is extremely low, and the value of Q2 in the permutation test turned out to be higher. We deeply thank the reviewer for the insightful comment. The comment is correct. We actually did not use the plasma (-) dataset for the further processing of our data as it exhibited low predicted ability, but mistakenly we had not pointed out in the manuscript. This has been clearly indicated in our revised version.

Line 433: the authors report that samples (plasma, kidney and urine) were collected during and at the end of the experiments. I didn't realize at what time-point each of the samples was collected. I thought they were all collected at the end of the experiment. Can you please clarify. Thank you again for this comment which aids us towards improving our text. We actually collected the data only at the end and not during the experiment. We have now clarified it In the revised version.

Sometimes authors refer to the three samples as tissues (example line 463 and 525), which is incorrect as only the kidney is a tissue. Urine and plasma are biofluids. So it will be more correct to treat the 3 by biological samples. Thank you for suggestion, we have replace the term tissues with the term biosamples in the new version of our text.

Line 484 and line 487: can you please clarify the difference between “hard” and “soft” homogenization? What are the experimental conditions for carrying out each of them? Thank you for pointing out this omission. The information regarding hard and soft module of homogenization have been clarified in our new version.

I think it would be extremely important to include the PCA with the projection of QCs for all datasets as a supplementary material. Thank you again for aiming to improve the manner we provide the information content of our study. We have included the PCA models of samples and QCs before and after the signal correction in the supplementary material.

Line 548: The authors refer that they tested different types of transformations. But it was not clear whether the final data was transformed or not? If yes, what method was applied? Thank you very much for pointing this out. We performed several types of transformations (log, cube and 1/sqrt) but they resulted in no improvement of the normality, so finally, the data were not transformed. This is now clarified in our revised text.

Line 548-549: Different scaling methods emphasizes and suppresses different kinds of features. Please specify and discuss how UV scaling affected your data. Thank you once again for the very interesting comment. Although the reviewer is right, it is difficult to evaluate the impact of the scaling for each feature, our criterion was to maximize the discrimination between our experimental groups.

Table 5: How were the pre-processing parameter determined or optimized?

According to the requirements of each analysis, we have empirically estimated the noise level and the base line threshold for the mass detection, the chromatogram building and the chromatogram deconvolution.